# Cross-Cultural Adaptation and Validation of the Fear of COVID-19 Scale for Chinese University Students: A Cross-Sectional Study

**DOI:** 10.3390/ijerph19148624

**Published:** 2022-07-15

**Authors:** Wanqiu Yang, Peng Li, Yubo Huang, Xiao Yang, Wei Mu, Wangwei Jing, Xiaohong Ma, Xiangyang Zhang

**Affiliations:** 1The Mental Health Center, Yunnan University, Kunming 650091, China; yangwanqiu@ynu.edu.cn (W.Y.); benjamin@mail.ynu.edu.cn (Y.H.); muwei1030@ynu.edu.cn (W.M.); 2Faculty of Education, Yunnan Normal University, Kunming 650500, China; lee@ynnu.edu.cn; 3Kunming Psychiatric Hospital, Kunming 650225, China; yangxiao@wchscu.cn (X.Y.); seaky194124@163.com (W.J.); 4West China Hospital of Sichuan University, Chengdu 610044, China; 5CAS Key Laboratory of Mental Health, Institute of Psychology, Chinese Academy of Sciences, Beijing 100101, China

**Keywords:** COVID-19, the Fear of COVID-19 Scale, reliability, validity, Chinese university students

## Abstract

Background: fear of COVID-19 is widespread among the population, especially among college students because of their increased exposure to the media information overload of the COVID-19 outbreak. The Fear of COVID-19 scale (FCV-19 S) is a relatively short instrument used to evaluate fears surrounding the COVID-19 pandemic. However, the validity and reliability of the Fear of COVID-19 Scale have not been fully investigated in Chinese university student groups. Objectives: this study assessed the cross-cultural adaptability and reliability of the FCV-19S for Chinese university students. Methods: a Chinese version of Fear of COVID-19 Scale (C-FCV-19S) was generated using the translation-backward translation method. Psychometric properties of the C-FCV-19S, including internal consistency, split-half reliability, construct reliability, convergent validity, and diagnostic accuracy, were evaluated. The Patient Health Questionnaire (PHQ) and Generalized Anxiety Disorder Questionnaire (GAD-7) scales were also used to evaluate participants for depression and anxiety. Results: the C-FCV-19S has acceptable internal consistency (Cronbach’s alpha: 0.872) and satisfactory split-half reliability (correlation coefficient: 0.799). Using the exploratory factor analysis (EFA), we examined the construct reliability (KMO = 0.920). The confirmatory factor analysis (CFA) confirmed that the bifactor model of scale (including general factor, factor1: the awareness of COVID-19 and physiological arousal, factor 2: fear-related thinking) had a good fit index (χ2/df =6.18, RMSEA= 0.067, SRMR = 0.028, GFI = 0.986, TLI = 0.970 and CFI= 0.988). Using depression-positive and anxiety-positive scores as reference criteria, we found that the areas under the curve were 0.70 and 0.68, respectively, and that the optimal cutoff scores of the C-FCV-19S was 17.5 (sensitivity: 66.3% and 58.7%, respectively). Conclusions: the validity and reliability of C-FCV-19S are satisfactory, and the optimal cutoff point was 17.5. The C-FCV-19S can be applied adopted in Chinese university students.

## 1. Introduction

The Coronavirus pandemic 2019 (COVID-19) spread rapidly all over the world and has posed serious public health challenges worldwide [1,2]. The public has been experiencing not only physical health problems but also psychological crisis [3,4]. As the number of infectious cases and mortality rate rapidly increased, fear of COVID-19 leads to individual-level mental distress such as anxiety, depression, post-traumatic stress disorder, and suicide, but also some psychosocial problems, including stigmatization, discrimination, disruption of community interactions [5,6]. Therefore, screening public fears related to COVID-19 and then taking further intervention will be beneficial for both preventing psychosocial problems and carrying out governmental public health interventions.

The Fear of COVID-19 Scale (FCV-19S) was developed in an Iranian context in 2020 and shown to have strong reliability and validity scale for assessing fears related to the coronavirus [7]. The final version of FCV-19S was a single-dimensional scale with 7 items and was shown to be significantly correlated with depression and anxiety, making it helpful for identifying these comorbid disorders [7,8,9]. Subsequently, the FCV-19S has been translated into eighteen different languages [8]. Most of these studies showed that it is a unidimensional scale. However, studies proposed a two-factor structure, such as in Israeli sample [10], Ecuadorian sample [11], Chinese population sample [12] and Russian adolescents [13]. These inconsistent results also show that it is an unstable factor structure of the FCV-19 Scale [14]. Meantime, most studies were small or middle-aged samples. These included work in Iranian (N = 717, mean age: 31) [7], Italian (N = 249, mean age: 34) [14], Saudi (N = 639, mean age: 35) populations [15]. These discrepancies may be due to differences in sample characteristics, cultural backgrounds, or experiences of the COVID-19 epidemic, including different, countries, ethnic groups, epidemic control situations, and so on. Therefore, the FCV-19 scale’s psychometric properties should be further studied across different cultures or vulnerable samples, especially in the elderly, adolescents, and clinical samples [16,17].

In China, approximately 80,000 individuals have been diagnosed with COVID-19, with over 4600 officially recorded deaths (Chinese National Health Commission 2020). The massive infectious public health event has put enormous pressure on the Chinese government, health care providers, and the public [18]. Level 1 public health response was activated in 31 Chinese provinces [19]. There are 33.66 million college students nationwide, including 8.83 million inter-provincial students. The continuous spread of the epidemic, strict isolation measures and delays in starting schools, colleges, and universities across the country were expected to influence the mental health of college students [20,21]. Existing studies have found that university students were more vulnerable to the harmful effects of media information overload of the COVID-19 outbreak, including panic, anxiety, and depression [22,23,24]. Incidences of anxiety and depression among Chinese university students were up to 40–50% during the COVID-19 epidemic [1,25,26,27]. One of the responsibilities of universities is to protect the physical and mental health of students and prevent the possible consequences of the spread of the epidemic [28]. During the COVID-19 epidemic, the National Health Commission of China issued a number of measures to reduce the spread of the virus, such as lockdowns, quarantine, and online teaching, and implemented emergency psychological crisis intervention for the public [29]. Meantime, the Ministry of Education of China has issued guidelines on mental health services in universities, including screening, monitoring the mental health status, and increasing the number of full-time and part-time psychological counselors [30].

Although the epidemic has been well under control in China, sporadic outbreaks have still occurred with most of the infected cases arising from southwest border cities or villages. Therefore, the present study will cross-culturally adapt and validate the FVC-19S in university students, which will contribute to mental health care in universities.

## 2. Materials and Methods

### 2.1. Study Design and Participant Characteristics

The cross-sectional study was carried out using a professional online survey in southwest China, from October to November 2020. We used an online survey website (www.wjx.cn) to create and distribute the survey. The data was collected in the classroom by investigators from four universities, including two key universities (excellent universities) and two common Universities. A total of 2550 participants were studying in the universities and 2334 of which were valid (effective rate = 91%). According to the principle of sample size, the ideal sample size of the preliminary and final survey should be 5–10 times and 40–50 times those included in the questionnaire [31]. The questionnaire included 26 items (they sum the number of the items for each instrument), thus meeting the ideal sample size. Inclusion criteria were as follows: undergraduates (age 18–26 years old) in southwest China took part in the study voluntarily. We randomly divided the 2334 samples into two equally-sized samples (N = 1167): sample 1 and sample 2.

Overall, 2334 individuals were included, among which 965 and 1369 students were from key and common universities, respectively. Their average age was 19 ± 1.29 years (range: 17–26). A total of 1275 students (54.6%) were male, and 1059 students (45.4%) were female. Means scores on the C-FCV-19S, PHQ-9 and GAD-7 were 16.04 ± 6.12, 5 ± 4.82, and 3 ± 3.98, respectively. Further details are presented in Table 1.

### 2.2. Ethical Considerations

The study was approved on 3 June 2019 by the corresponding Institutional Review Board (IRB) of the Institute of Psychology, and the Chinese Academy of Sciences. Electronic informed consent was obtained before data were collected.

### 2.3. Measures

#### 2.3.1. Fear of COVID-19 Scale (FCV-19S)

Daniel Kwasi Ahorsu’s research team developed the Fear of COVID-19 Scale (FCV-19S) in March 2020. The scale was tested in the general Iranian population (*n* = 717 Iranian) and was shown to have good internal consistency (α = 0.82) and concurrent validity (r = 0.425–0.511). The original FCV-19S (English version) included 10 items. It was later revised into a 7-item unidimensional scale with a five-point Likert rating scale, with 1 corresponding to strongly disagree and 5 corresponding to strongly disagree 5. The overall score indicates a level of fear, with a higher score indicating a higher degree of fear.

In this study, we took the following steps to create a Chinese version of the FCV-19S. First, researchers revised the original of 10-item FCV-19S scale and decided that the original version was both more comprehensive and culturally suitable. Second, the 10-item FCV-19S was translated into Chinese by three linguistic experts. Then, the researchers examined the translated items. After that, another linguistic expert completed a back-translation. These two translation processes ensured that the content of the 10 items reflected in the translation was the same as that in the original version. The Chinese version of the FCV-19S (C-FCV-19S) was also applied to 15 students as a pilot test to collect information and ensure that each item was understood accurately. After minor revision, the C-FCV-19S was finalized and consisted of 10 items that were rated on a five-point Likert scale.

#### 2.3.2. Patient Health Questionnaire 9 (PHQ-9)

The PHQ scale is a useful measure for assessing depression severity with good validity and reliability. Respondents reported the presence of each symptom within the last 2 weeks. The scale is a 4-point Likert scale (ranging from 0—“not at all to” to 3—“nearly every day”). Higher overall scores on the PHQ-9 indicate more severe general anxiety, with scores of 1–4 = minimal, 5–9 = mild, 10–14 = moderate, 15–19 = Moderately severe, and 20–27 = Severe [32]. In this study, the Cronbach’s α of PHQ-9 in this sample was 0.908. A cutoff point of 9 was used to divide participants into groups with or without moderate depression.

#### 2.3.3. Generalized Anxiety Disorder Questionnaire (GAD-7)

The GAD-7 scale is used to screen for generalized anxiety disorder (GAD) and to assess its severity in clinical practice. Respondents report the presence of each symptom within the last 2 weeks, using a 4-point Likert scale (ranging from 0—“not at all to” to 3—“nearly every day”). Higher overall GAD-7 scores indicate more severe general anxiety, with scores of 0–4 = minimal, 5–9 = mild, 10–14 = moderate, and 15–27 = Severe [33]. In this study, the Cronbach’s α of GAD-7 in this sample was 0.923. A cutoff point of 9 was used to divide participants into groups with or without moderate anxiety.

### 2.4. Statistics Analysis

#### 2.4.1. Reliability

Each item of the C-FCV-19S was selected and validated based on classical test theory (CTT). Items with a standard deviation greater than 1 were collected to evaluate sensitivity. We correlated each item in the C-FV-19S with the total score to evaluate internal consistency. Items were removed from the analysis if the item-total correlation (CITC) was ≤0.40 [34]. A Cronbach’s alpha greater than 0.80 signified good internal consistency [35]. The split-half reliability of this scale was also evaluated.

#### 2.4.2. Validity

After the item-level analysis, an exploratory factor analysis (EFA) to explore the dimensions of the questionnaire. The EFA was completed by using SPSS (version 26.0) on sample 1. Two or more Eigenvalues greater than 1 would have indicated that the scale was not unidimensional. Items with factor loading less than 0.4 were excluded from the questionnaire [36]. Our construct reliability (CR ≥ 0.70), standardized factor loadings (ranging from 0.50–0.95; a significance level of 0.05), and average variance extracted (AVE ≥ 0.50) [37] suggested excellent convergent validity.

Following EFA, we used Confirmatory Factor Analysis (CFA) to analyze the validity of the model structure. CFA was carried out using AMOS (version 17.0) on sample 2. The CFA model fix index includes: ratios [38], the root mean square error of approximation (RMSEA) [39], standardized root mean square residual (SRMR) [40], comparative fit index (CFI) [40], Tucker–Lewis index (TLI) [41] and GFI [40] index. Although a simple rule of thumb suggests that a χ^2^/df ratios less than 3 means that the model’s fit is acceptable, some researchers have proposed that this is not the case due to different sample sizes and test lengths [42]. An RMSEA value is less than 0.05, the model can be assumed to demonstrate a “good” fit and “acceptable” fit to the data, respectively [39]. A cutoff value close to 0.08 for SRMR implies a relatively good fit [40]. CFI, TLI, or GFI greater than 0.09 corresponds to an “acceptable” fit [39].

The convergent validity of the C-FV-19S was examined through correlation analysis of the C-FV-19S and the PHQ-9 and GAD-7. The correlation values for convergent validity were categorized as follows: weak correlation (0.10–0.29), moderate correlation (0.30–0.49), and strong correlation (0.50–1.0) [43].

## 3. Results

### 3.1. Item Analysis

The total score of the scale (10 items) for 2334 participants was analyzed, and all scores were ranked in ascending order. Participants ranked in the bottom 27% and the top 27% were defined as having low and high fear of COVID-19, respectively. The *t*-values of the low and high fear groups were 9.373 ± 1.34 and 23.64 ± 4.27, respectively, and an independent samples *t*-test showed they were significantly different (*p* < 0.001). The total score and the score of each item both passed the Kolmogorov–Smirnov normality test (*p* < 0.05), suggesting a correlation between each item and the total score. These correlations were further examined using Spearman’s rank correlation (shown in Table 2) and ranged from 0.636 to 0.783.

### 3.2. Reliability Analysis

Internal consistency was evaluated using Cronbach’s alpha and split-half reliability. Cronbach’s alpha coefficient was 0.872, indicating a high level of internal consistency. The coefficient of the split-half reliability was 0.799.

### 3.3. Validity Analysis

#### 3.3.1. Exploratory Factor Analysis (EFA)

We randomly divided the data into two halves and ran EFA on sample 1. The 10 items passed Bartlett’s test of sphericity and a Kaiser–Meyer–Olkintest applied to (χ^2^ = 8684.303, *p* < 0.001, KMO = 0.920), indicating that EFA was appropriate [44]. Since each item of the C-FCV-19S was interrelated, we employed a principal component analysis with Promax oblique rotation. All items’ factor loadings that were more than 0.50 met the requirements of two dimensions to ensure that the eigenvalues were above 1. Following the factor analysis, we obtained a two-factor scale with 10 items and determined that the cumulative variance contribution rate of the two factors was 73.138%. The variance contribution rates of Factor 1 (named: awareness of the COVID-19 and physiological arousal) and Factor 2 (named: fear-related thinking) were 60.504% and 12.634%, respectively. The factor loadings of all items were greater than 0.50, indicating that the questionnaire had good structural validity. The factor loadings of items 7 and 8 exceeded 0.5 on both Factor 1 and Factor 2. Thus, these two items were excluded, leaving eight items (see Appendix A). The scree plot of the last EFA and the results of the EFA are shown in Figure 1 and Table 3, respectively.

#### 3.3.2. Confirmatory Factor Analysis (CFA)

We used the Maximum Likelihood Estimates in the CFA and ran CFA on sample 2. Model 1 was a unidimensional model, Model 2 was a two-factor model, and Model 3 was a bifactor model. The fit indices of the two models are shown in Table 4. The model fit of the single-factor model (Model 1) was unqualified as its RMSEA values (0.229) and χ^2^/df (61.95), the SRMR (0.1002) was higher than the ideal value (≤0.08) and the GFI (0.754), TLI (0.946), and CFI (0.747) indices were also lower than the ideal value (0.90). For Model 2, the RMSEA values (0.140) and χ^2^/df (23.68) were above the ideal value (≤0.08 and ≤5), and the SRMR (0.0774) was close to the ideal value (≤0.08). The GFI (0.907), TLI (0.906), and CFI (0.936) indices were also close to the ideal value (0.90). The fit indices of Model 2 were better than Model 1, but some indicators still fall short of the ideal values. For Model 3, although χ_2_/df (6.18) was just a little higher than the ideal value (≤5), the RMSEA values (0.067) and the SRMR (0.028) that was obtained to the ideal value (≤0.08). The GFI (0.986), TLI (0.970), and CFI (0.988) indices were also obtained to the ideal value (0.90) (shown in Table 4 and Figure 2).

#### 3.3.3. Convergent Validity

Convergent validity was confirmed by correlation analysis between the C-FCV-19S and PHQ-9 or GAD-7, using Spearman’s correlation coefficients. There were significant correlations between the total scores on the C-FCV-19S and the total scores on the HPQ-9 (r = 0.326, *p* value < 0.001) and GAD-7 (r = 0.346, *p* value < 0.001) scales.

#### 3.3.4. Diagnostic Accuracy and Criterion Validity

The results of the ROC analysis were compared to the depression-positive dimension (shown in Figure 3). The area under the curve (AOC) of the C-FCV-19S was 0.68 (95% confidence interval [CI] 0.652–0.709). The sensitivity and specificity of the anxiety-depression dimension, classified using the C-FCV-19S, varied considerably as a function of different cut-off points. When the cutoff point was increased from 8.5 to 27.5, the sensitivity decreased from 94.4% to 11.4%, and the specificity increased from 12.3% to 97.5%. Based on the maximum value of the Youden Index (0.299), the best compromise between sensitivity (58.7%) and specificity (71.2%) was, again, a cutoff point at 17.5.

The results of the receiver operating characteristic curve (ROC) analysis, which we compared to the anxiety-positive dimensions as a reference standard, are shown in Figure 4. The AOC of the C-FCV-19S was 0.70 (95% CI: 0.662–0.744). When the cutoff point was increased from 8.5 to 29.5, the sensitivity decreased from 94.6% to 11.4%, and the specificity increased from 11.4% to 97.9%. Based on the maximum value of the Youden Index (0.343), the best compromise between sensitivity (66.3%) and specificity (68%) was obtained at a cutoff point of 17.5.

Furthermore, participants were divided into the without and with fear COVID-19 groups according to the cutoff point of 17.5. As shown in Table 5, compared to the without fear of COVID-19 group, the group with of fear COVID-19 had significantly higher HAMA scores (2.51 ± 3.38 vs. 4.80 ± 4.56, *p* < 0.001), HAMD scores (3.98 ± 4.17 vs. 6.56 ± 5.45, *p* < 0.001), and higher rates of anxiety and depression (all *p* < 0.001) symptoms. Further binary logistic regression was used to predict anxiety (B = 0.119, df = 1, OR = 1.126; 95%CI: 1.102–1.151; *p* < 0.001) and depression symptoms (B = 0.110, df = 1, OR = 1.116; 95%CI: 1.097–1.135; *p* < 0.001).

## 4. Discussion

This study assessed the cultural adaptation of the C-FCV-19S and found it to be psychometrically valid for Chinese university students and to help quickly screen the fear of COVID-19. This study had three principal findings. (1) The finalized C-FCV-19S had two dimensions and 8 items. (2) The C- FCV-19S had good reliability and validity, with an optimal cutoff point of 17.5. (3) The C-FCV-19S score was a positive association with anxiety and depressive symptoms.

Ahorsu et al. developed the FCV-19s as a timely self-evaluation measurement meant to assess fear of COVID-19 during the pandemic. The final version of the FCV-19s was a unidimensional scale with 7-item [7]. Other research in different countries validated the scale and had inconsistent results. Studies in Italian and Arabic general populations [14,15] and Spanish university students also demonstrated that the 7-item version had a unidimensional structure [45]. However, another study, such as in Japanese adolescents [46] and university students of Russia [13] showed a bi-factor scale. It is worth noting that two studies for the Chinese sample so far were inconsistent. Specifically, one study of Chinese students, including schools and universities reported a single-dimensional structure scale [9], while another study of the general Chinese population found a two-dimensional structure scale [12]. However, our study found that the 8-item C-FCV-19s was a bi-factor model. One reason for these differences may be sample characteristics and size. We collected a large sample of university students (N = 2334), whose average age was 19 years old (range: 18–26), while other studies of the Chinese population used teenagers and adults (N = 1700; mean age: 18; range: 10–57) [12]. In the present study, the two deleted items were “I am afraid of losing my life because of Coronavirus-19” and “When watching news and stories about Coronavirus-19 on social media, I become nervous or anxious”. These descriptions may not fit well with the current situation of the epidemic in China, as the COVID-19 epidemic has overall been well-controlled and Chinese people tend to believe in the government’s ability to deal with COVID-19 [47]. More importantly, this difference may come from the specificity of college students as a group. It has been illustrated that young people with higher levels of moral disengagement are less likely to engage in preventive behaviors in public crisis situations such as the COVID-19 pandemic [48,49]. In addition, 7 and 8 items of the FCV-19S belong to threat perception. It seems that in young people higher levels of perceived threats concerning personal health or the health of loved ones did not correspond to a greater probability to adopt preventive behaviors [50]. One possible explanation is that adolescents are at a much lower risk of contracting the most severe symptoms of COVID-19 [51] and therefore they consider the probability of being infected with serious harm to be very low. In other words, threat perception does not influence the adoption of healthy behaviors by adolescents in the face of the COVID-19 epidemic [50]. Taken together, we have good reason to speculate that two items are excluded from the CFA analysis.

Our study supported the idea of satisfactory reliability and validity of the two-dimension structure of the C-FCV-19S. Firstly, EPA found that two factors of the C-FCV-19S, based on factors with eigenvalues, were greater than those of random datasets. Further, CFA suggested that the C-FVC-19S was consistent with the data, which corroborates previous research indicating that the data model fits the two-factor structure. In the SEM, the model fit GFI, TLI, and CFI indices were all obtained to the ideal value (*n* > 1000), although the RMSEA values and χ^2^/df were above the ideal value. Some studies have suggested that this was acceptable in large sample studies (*n* > 1000) [40,52]. Traditionally, the accepted standard of fit indices (GFI, TLI, CFI, etc.) > 0.9 or above to be acceptable, and models with RMSEA between 0.05 and 0.08 are acceptable [41]. However, recent research suggests that the analysis of these indices was based on MLE (Maximum Likelihood Estimate) and CLS (Generalized Least Squares), and it was recommended that SRMR should be combined with TLI, RNI, or CFI indicators to test SEM fits [40,53]. Meantime, if the sample is above 1000 (N > 1000), the chi-square value (χ^2^) is usually so large that models with good fit are rejected. The index of χ^2^/df is used when the sample is less than 1000 (N < 1000) [41,54]. Therefore, there is good reason to conclude that our model has the goodness of model fit to support the two-factor of the C-FCV-19S.

Unfortunately, to date, only a study reported that the cutoff scores of the Greek FCV-19S version were 16.5 or higher for distinguishing elevated fear and normal of COVID-19 [8]. We used a ROC analysis to determine the optimal cutoff point of 17.5. Most studies of FCV-19s in different countries showed that the mean score of FCV-19s was close to 17.5, with ranged from 15.6 to 18.3 [11,16,45,55]. Among them, the mean score of Russian university students was 18 ± 4.5 [13]. Thus, these studies illustrated that the cutoff score of our study is reasonable and credible.

This study has several limitations that should be noted. First, the representativeness of the sample was a limitation. We only surveyed university students in southwest China, where COVID-19 was most prevalent. Second, confounding factors, including students’ hometowns, family health, and disease history should be fully considered. Finally, some participants may have had depression and anxiety prior to the COVID-19 pandemic, which may have influenced their fear of COVID-19. However, we were unable to collect this information in the present cross-sectional study. Therefore, the criterion validity of the C-FVS scale in this study has yet to be verified.

## 5. Conclusions

In summary, the psychometric validation of the C-FCV-19S (8-item version) is the first in Chinese university students. The scale shows to be reliable and valid in the measurement of fear of COVID-19 and provides a simple and rapid solution to screening for fear of COVID-19. The optimal cutoff point of the scale was 17.5 for Chinese university students. It might be beneficial for the development of interventions for fear related to the COVID-19, as well as providing mental health services in universities.

## Figures and Tables

**Figure 1 ijerph-19-08624-f001:**
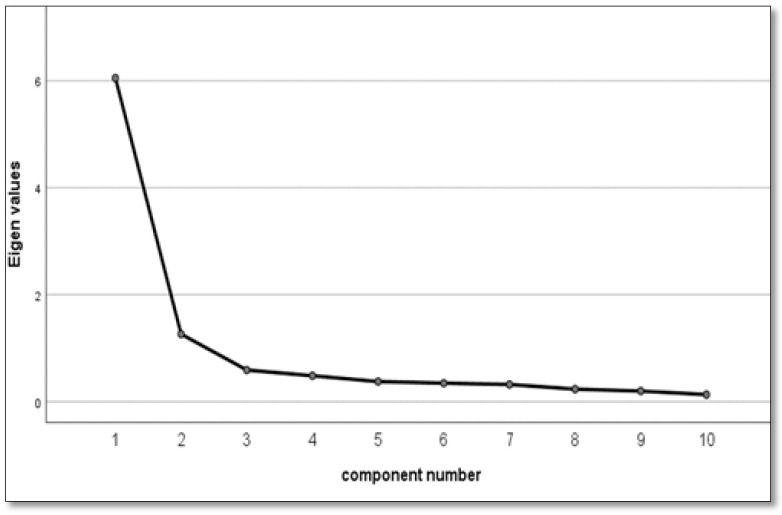
Scree Plot based in EFA of the FCV-19S.

**Figure 2 ijerph-19-08624-f002:**
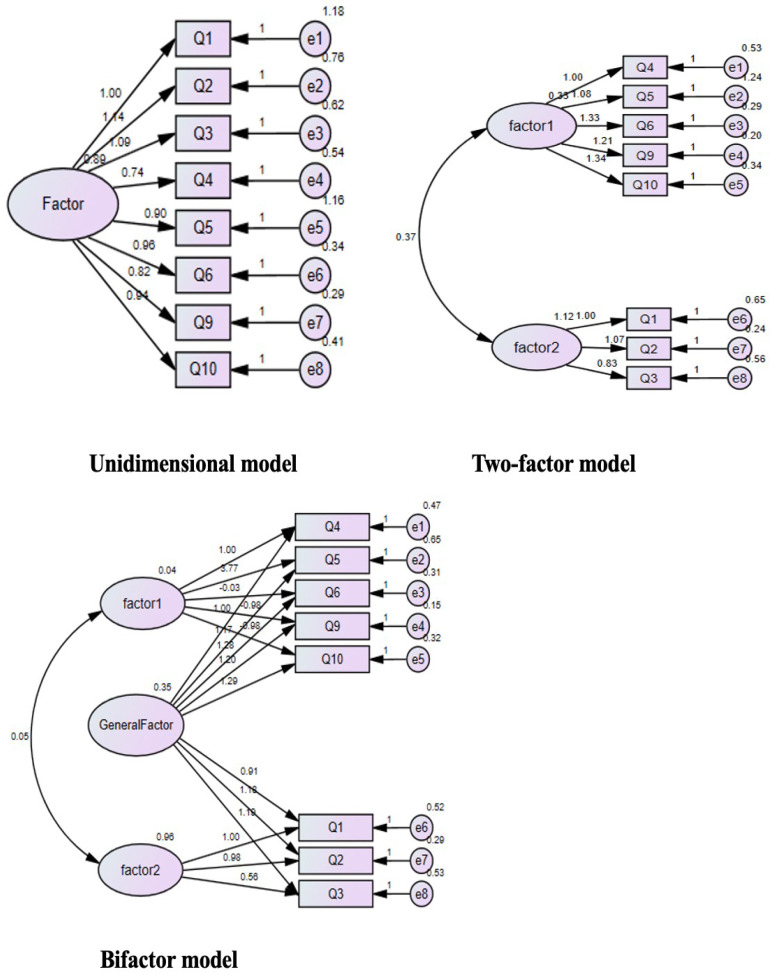
Graphical representation of the three compared measurement models: (1) unidimensional model; (2) two-factor model; (3) bifactor model (factor 1: the awareness of COVID-19 and physiological arousal. factor 2: fear-related thinking).

**Figure 3 ijerph-19-08624-f003:**
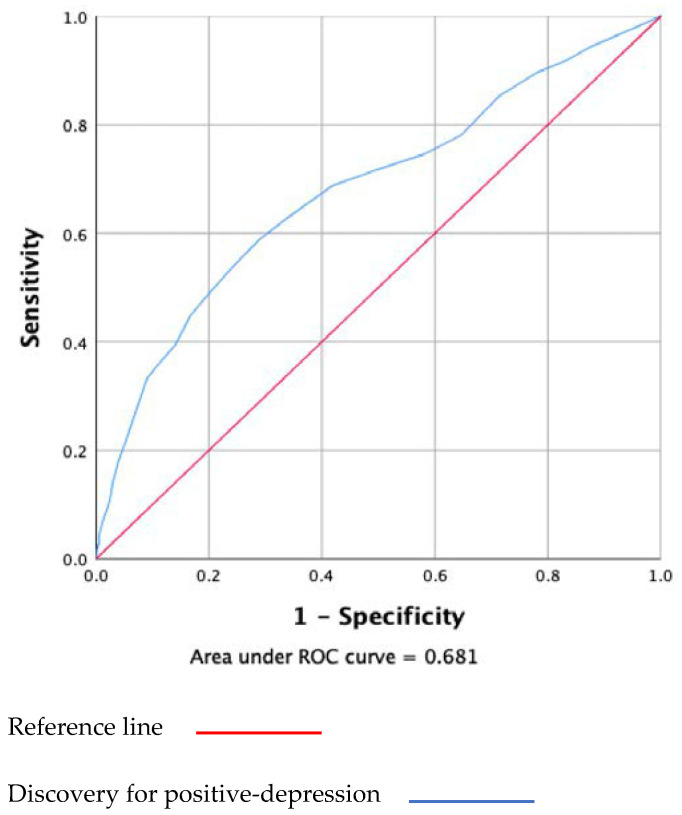
The receiver operator curve (POC) of the C-FCV-19S based on positive-depression.

**Figure 4 ijerph-19-08624-f004:**
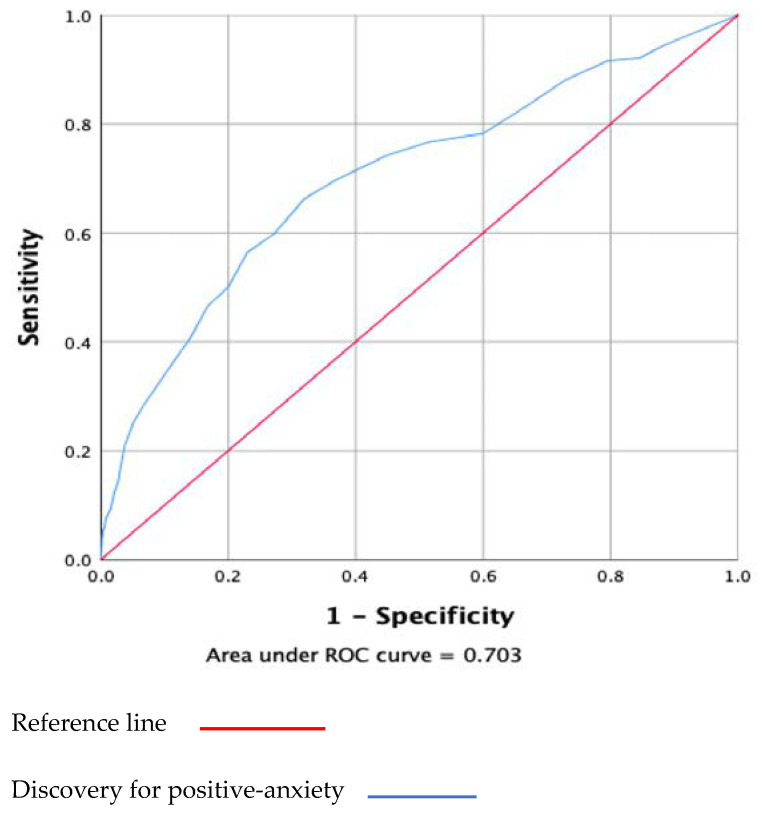
The receiver operator curve (POC) of the C-FCV-19S based on positive-anxiety.

**Table 1 ijerph-19-08624-t001:** Participant characteristics (*n* = 2234).

Variable		Mean ± Standard Deviation or *n* (%)
Age (years old)		19 ± 1.29
	17–19	1075 (46.1)
	20–22	1198 (51.3)
	23–29	61 (2.6)
University		
	key university	965 (41.3)
	common university	1369 (58.7)
Gender		
	Male	1275 (54.6)
	female	1059 (45.4)
PHQ-9		4.88 ± 4.82
	Negative-depression	1850 (79.3)
	Positive-depression	484 (20.7)
GAD-7		3.31 ± 3.98
	Negative-anxiety	2132 (91.3)
	Positive-anxiety	202 (8.7)
C-FV-19S		16.04 ± 6.12 (8–50)

**Table 2 ijerph-19-08624-t002:** Item Analysis of the Chinese version Fear of COVID-19 scale (The C-FCV-19S).

The C-FCV-19S Items	Mean (SD)	Skewness	Kurtosis	Correlation	*p*-Value
1. I am most afraid of corona virus-19.	3.02 (1.33)	−0.037	−1.134	0.678	<0.001
2. It makes me uncomfortable to think about coronavirus-19.	2.61 (1.24)	0.314	−0.888	0.783	<0.001
3. I worry a lot about coronavirus-19.	2.30 (1.15)	0.577	−0.500	0.771	<0.001
4. Coronavirus-19 is almost always terminal.	1.61 (0.93)	1.503	1.671	0.64	<0.001
5. Coronavirus-19 is an unpredictable disease.	2.36 (1.28)	0.551	−0.786	0.636	<0.001
6. My hands become clammy when I think about coronavirus-19.	1.60 (0.936)	1.646	2.274	0.772	<0.001
7. I am afraid of losing my life because of coronavirus-19.	2.15 (1.30)	0.858	−0.424	0.747	<0.001
8. When watching news and stories about coronavirus-19 on social media, I become nervous or anxious.	2.28 (1.18)	0.562	−0.680	0.765	<0.001
9. I cannot sleep because I’m worrying about getting coronavirus-19.	1.46 (0.83)	1.895	3.225	0.715	<0.001
10. My heart races or palpitates when I think about getting coronavirus-19.	1.60 (0.97)	1.660	2.116	0.741	<0.001

**Table 3 ijerph-19-08624-t003:** Factor loadings of 10 items in test sample 1 (N = 1167).

Items	Factor 1	Factor 2
Q9. I cannot sleep because I’m worrying about getting coronavirus-19.	0.893	0.209
Q10. My heart races or palpitates when I think about getting coronavirus-19.	0.861	0.255
Q6. My hands become clammy when I think about coronavirus-19.	0.851	0.314
Q 4. Coronavirus-19 is almost always terminal.	0.82	0.169
Q5. Coronavirus-19 is an unpredictable disease.	0.584	0.402
Q7. I am afraid of losing my life because of coronavirus-19.	0.575	0.544
Q8. When watching news and stories about coronavirus-19 on social media, I become nervous or anxious.	0.571	0.578
Q1. I am most afraid of corona virus-19.	0.076	0.904
Q2. It makes me uncomfortable to think about coronavirus-19.	0.288	0.869
Q3. I worry a lot about coronavirus-19.	0.452	0.698
Eigen values	6.05	1.263
Variance contribution rate	60.5	12.63
Cumulative variance contribution rate (%)	60.5	73.14

**Table 4 ijerph-19-08624-t004:** Fit indices of models in the CFA of the Fear of COVID-19 Scale in test sample 2.

	χ^2^ (df.)	χ^2^/df	RMSEA	SRMR	GFI	CFI	TLI	NFI	RFI	IFI
Threshold value		≤5.0	≤0.08	≤0.08	>0.09	>0.09	>0.09	>0.09	>0.09	>0.09
Model 1	1239.172 (20) ***	61.95	0.229	0.1002	0.754	0.747	0.646	0.744	0.642	0.748
Model 2	450.63 (19) ***	23.68	0.14	0.0774	0.907	0.936	0.906	0.934	0.902	0.936
Model 3	68.055 (11) ***	6.18	0.067	0.028	0.986	0.988	0.970	0.986	0.964	0.988

*** Indicates significance at the 0.001 level. Model 1: single-factor model; Model 2: two factors model consisting of factor 1 and factor 2. Model 3: bifactor model.

**Table 5 ijerph-19-08624-t005:** Comparison of with and without fear COVID-19 in anxiety and depression.

		Without Fear COVID-19	With-Fear COVID-19	F/χ^2^	*p*-Value
		*n* = 1518	*n* = 682		
Anxiety		2.51 ± 3.38	4.80 ± 4.56	189.562	<0.001
	without	1450 (68)	682 (32)	95.737	<0.001
	with	68 (33.7)	134 (66.3)		
Depression		3.98 ± 4.17	6.56 ± 5.45	163.798	<0.001
	without	1318 (71.2)	532 (28.8)	151.045	<0.001
	with	200 (41.3)	284 (58.7)		

## Data Availability

The datasets used and analyzed in the current study are available from the corresponding author upon reasonable request.

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
