# Peer review of "Cross-Cultural Adaptation and Validation of the Fear of COVID-19 Scale for Chinese University Students: A Cross-Sectional Study"

_ijerph, 2022, doi:10.3390/ijerph19148624_

Round 1
Reviewer 1 Report
Congratulations for this paper. I will add some comments:
- it is very important that this study has ethical considerations
- study took all steps necessary to psychometric validation
- I don´t understand this phrase: Exiting studies have reported a positive correlation between mobile phone use and
psychological stress during the COVID-19 epidemic [16, 17]. Is it an goals of the study? Have been measured in this study the use of phone? Please, review - In Spanish university students were made a similar study, too. In this study, authors had an unidimensional structure. Besides the mean score was 16.79.
Author Response
Response to Reviewer 1 Comments
Point 1: Congratulations for this paper. I will add some comments: it is very important that this study has ethical considerations study took all steps necessary to psychometric validation. I don´t understand this phrase: Exiting studies have reported a positive correlation between mobile phone use and psychological stress during the COVID-19 epidemic [16, 17]. Is it an goals of the study? Have been measured in this study the use of phone?
Response 1:
Thank you very much for your careful review. In our study, it was not the goal of the study to explore the relationship between mobile phone use and psychological stress during the COVID-19 epidemic. The use of phones hasn't been measured in this study. Therefore, we have deleted the sentence and revised it in the Introduction section on Page 2 Lines 74-76, showing as follows: “Existing studies have found that university students were more vulnerable to the harmful effects of media information overload of the COVID-19 outbreak, including panic, anxiety, and depression [22-24]”.
Point 2: Please, review: In Spanish university students were made a similar study, too. In this study, authors had an unidimensional structure. Besides the mean score was 16.79.
Response 2:
Thank you very much for your excellent suggestions. We have downloaded the paper and cited it as a reference in the revised version on Page 10 Lines 287-289, showing as follows: “Studies in Italian and Arabic general populations and Spanish university students also demonstrated that the 7-item version had a unidimensional structure”.
Reference : M Martínez-Lorca, A Martínez-Lorca, Criado-Lvarez J J , et al. The Fear of COVID-19 Scale: Validation in Spanish university students[J]. Psychiatry Research, 2020, 293:113350. DOI:10.1016/j.psychres.2020.113350

Reviewer 2 Report
Discusssion section can be revised to be less statistics-centered. Conceptual and methodological discussions of the findings can be increased.
Author Response
Response to Reviewer 1 Comments
Point 1: Discusssion section can be revised to be less statistics-centered. Conceptual and methodological discussions of the findings can be increased.
Response 1:
Thank you very much for your excellent suggestion. We have added the discussion by looking at the cognitive characteristics of the university students and the situation of epidemic prevention and control in China in the Discussion section on Page 10 Lines 300-325, showing as follows: “ These descriptions may not fit well with the current situation of the epidemic in China, as the COVID-19 epidemic has overall been well-controlled and Chinese people tend to believe in the government’s ability to deal with COVID-19 [49]. More importantly, this difference may come from the specificity of college students as a group. It has been illustrated that young people with higher levels of moral disengagement are less likely to engage in preventive behaviors in public crisis situations such as the COVID-19 pandemic [50,51]. In addition, 7 and 8 items of the FCV-19S belong to threat perception. It seems that in young people higher levels of perceived threats concerning personal health or the health of loved ones did not correspond to a greater probability to adopt preventive behaviors [52]. One possible explanation is that adolescents are at a much lower risk of contracting the most severe symptoms of COVID-19 [53] and therefore they consider the probability of being infected with serious harm to be very low. In other words, threat perception does not influence the adoption of healthy behaviors by adolescents in the face of the COVID-19 epidemic [54]. Taken together, we have good reason to speculate that two items are excluded from the CFA analysis”.

Reviewer 3 Report
The objetive of the study is assessed the cross-cultural adaptability and reliability of the FCV-19S for Chinese university students.
The FCV-19S is a scale already validated with the chinese population sample (Chi, X.; Chen, S; Chen, Y.; Chen, D.; Yu, Q.; Guo, T.; Cao, Q.; Zheng, X.; Huang, S.; Hossain, M.M.; Stubbs, B.; Yeung, A.; Zou, 358 L.Psychometric Evaluation of the Fear of COVID-19 Scale Among Chinese Population. Int J Ment Health Addict.2021,1-16).
The authors justify that the study is important to confirm the unidimensional structure of the scale, in addition to verifying the specificity of the population. However, they do not describe the total universe of students, nor the characteristics of this population (whether they were going to university at the time of data collection or not, whether they were in remote learning). These conditions may have influenced the results, since the authors themselves emphasize that the data collection was in a moment of well under control in China.
Other characteristics of the sample, such as, the fact that individuals could already have depression and anxiety problems prior to the pandemic, should have been controlled (this is a limitation of the study). An analysis by age is also suggested, since the sample was between 18 and 26 years old, a range that may be relevant in the analyses.
The statistical analyzes performed are adequate and consistent. In Figure 2 it is suggested to include the error variances between the items, since it can affect the indices of the RMSEA model, that are low.
Regarding to the statement, “These discrepancies may be due to differences in cultural backgrounds or experiences of the COVID-19 epidemic”. What other cultural characteristics might have influenced the results?
Author Response
Response to Reviewer 3 Comments
Point 1: The objetive of the study is assessed the cross-cultural adaptability and reliability of the FCV-19S for Chinese university students.
The FCV-19S is a scale already validated with the chinese population sample (Chi, X.; Chen, S; Chen, Y.; Chen, D.; Yu, Q.; Guo, T.; Cao, Q.; Zheng, X.; Huang, S.; Hossain, M.M.; Stubbs, B.; Yeung, A.; Zou, 358 L.Psychometric Evaluation of the Fear of COVID-19 Scale Among Chinese Population. Int J Ment Health Addict.2021,1-16).
The authors justify that the study is important to confirm the unidimensional structure of the scale, in addition to verifying the specificity of the population. However, they do not describe the total universe of students, nor the characteristics of this population (whether they were going to university at the time of data collection or not, whether they were in remote learning). These conditions may have influenced the results, since the authors themselves emphasize that the data collection was in a moment of well under control in China.
Other characteristics of the sample, such as, the fact that individuals could already have depression and anxiety problems prior to the pandemic, should have been controlled (this is a limitation of the study). An analysis by age is also suggested, since the sample was between 18 and 26 years old, a range that may be relevant in the analyses.
The statistical analyzes performed are adequate and consistent. In Figure 2 it is suggested to include the error variances between the items, since it can affect the indices of the RMSEA model, that are low.
Regarding to the statement, “These discrepancies may be due to differences in cultural backgrounds or experiences of the COVID-19 epidemic”. What other cultural characteristics might have influenced the results?
Response:
Thank you very much for your careful review. These points are very excellent. We have revised these issues in the revised manuscript, showing as follows:
First, we have added the characteristics of this population in the section of Study design and sample on page 2 Line 95-98, showing as follows: “The data was collected in the classroom by investigators from four universities, including two key universities (excellent universities) and two common Universities. A total of 2550 participants were studying in the universities and 2334 of which were valid (effective rate =91%).”
Secondly, some participants in this study may have had depression and anxiety prior to the pandemic, however, we were unable to obtain this information, which is one of the limitations of our study. Therefore, we have added this limitation in the discussion section on Page 14, Lines 343-346, showing as follows, “Finally, some participants may have had depression and anxiety prior to the CONVID-19 pandemic, which may have influenced their fear of CONVID-19. However, we were unable to collect this information in the present cross-sectional study ”.
Thirdly, according to your excellent suggestion, we analyzed the differences in the scores of anxiety, depression, and fear of COVID scores between the three groups aged 18-20, 21-23, and 24-26 years. However, there were no significant differences between the three groups.
Fourthly, according to your excellent suggestion, we have referred to some references for statistical analysis. and have analyzed the SEM again, by calculating the error variances between the items, the fit indices were better, but some indicators weren’t still close to the ideal values (χ2 /df =9.51, RMSEA= .085, SRMR =.05, GFI =.982, TLI =.951 and CFI=.984), shown as follows in Figure 1. we speculated there should be a hidden variable ( a general factor ). Then, we were future to explore the bifactor model, and the fit indices of the bifactor model were obtained to the ideal values (χ2 /df =6.18, RMSEA=.067, SRMR =.028, GFI =.986, TLI =.970 and CFI=.988). Therefore, we reported the bifactor model (Model 3).
(Figure 1)
Model 3: bifactor model
References:
[1] Leue A , Beauducel A . The PANAS structure revisited: on the validity of a bifactor model in community and forensic samples.[J]. Psychological Assessment, 2011, 23(1):215-25. 10.1037/a0021400
[2] Torres-Vallejos J , Juarros-Basterretxea J , Oyanedel J C , et al. A Bifactor Model of Subjective Well-Being at Personal, Community, and Country Levels: A Case With Three Latin-American Countries[J]. Frontiers in Psychology, 2021, 12. 10.3389/fpsyg.2021.641641
[3] Di M , Jia N , Wang Q , et al. A bifactor model of the Wong and Law Emotional Intelligence Scale and its association with subjective well-being[J]. The Journal of Positive Psychology, 2020(6):1-12. 10.1080/17439760.2020.1791947
Finally, the Fear of COVID-19 Scale has been validated by researchers from different countries and the findings are still inconsistent. We suggest that these differences may arise from the fact that different ethnic groups, countries, religions, and epidemic control situations all have an impact on the fear of COVID-19. Therefore, we revised the sentence on Page 10, lines 300-306, showing as follows, “ These descriptions may not fit well with the current situation of the epidemic in China, as the COVID-19 epidemic has overall been well-controlled and Chinese people tend to believe in the government’s ability to deal with COVID-19 [49]. More importantly, this difference may come from the specificity of college students as a group. It has been illustrated that young people with higher levels of moral disengagement are less likely to engage in preventive behaviors in public crisis situations such as the COVID-19 pandemic [50,51].”

Reviewer 4 Report
Dear Editor,
Thank you for giving me the opportunity to review the paper entitled "Cross-cultural Adaptation and Validation of the Fear of COVID-19 Scale for Chinese University Students: A cross-sectional study". I appreciated the content and originality of the contribution and I think it can be suitable for the publication to the Journal. Nevertheless, I have some concerns that I list below:
ABSTRACT
1) I found this sentence unclear, please revise. “However, the scale of further cross-cultural adaptation and validation in university students is needed.”
2) It would be helpful to add in the Abstract some information about the factorial structure of the scale (i.e. 2 dimensions, 8 items each).
INTRODUCTION
3) I think that the Introduction gives a concise description about the opportunity to validate the FCV-19S in the Chinese context. However I believe that some other data are needed.
4) The Authors should describe in more detail the original version of the scale (how many items, where it was developed, what are its psychometric properties, etc.) and the diverse adaptations. It could be also useful to add a table that summarize the different characteristics and properties of the diverse adaptations of the scale.
5) The Authors should also add a brief paragraph about Chinese context in which the study was conducted both in terms of the spread of the COVID-19 and of the policy adopted by the Authorities (e.g. restrictive policies to free movement of people, hygienic recommendations, etc). This could help the reader to understand the cultural, social and political background of the study.
6) I don’t understand why the Authors stated, “Exiting studies have reported a positive correlation between mobile phone use and psychological stress during the COVID-19 epidemic [16, 17].” Why have the Authors introduced the issue of the correlation between mobile phone use and psychological stress during the COVID-19? The link is unclear to me.
7) I think that the Authors should expand the description of the negative psychological effects of COVID for University students. It would be useful to add evidence from other studies conducted in China, as well as in other cultural contexts. Please, see the following articles for a non-exhaustive list.
Cao W, Fang Z, Hou G, Han M, Xu X, Dong J, Zheng J. The psychological impact of the COVID-19 epidemic on college students in China. Psychiatry research. 2020 May 1;287:112934. https://doi.org/10.1016/j.psychres.2020.112934
Ma Z, Zhao J, Li Y, Chen D, Wang T, Zhang Z, et al. Mental health problems and correlates among 746 217 college students during the coronavirus disease 2019 outbreak in China. Epidemiol Psychiatr Sci. (2020) 29:e181. doi: 10.1017/S2045796020000931
Wang C, Zhao H. The impact of COVID-19 on anxiety in Chinese university students. Front Psychol. (2020) 11:1168. doi: 10.3389/fpsyg.2020.01168
Son C, Hegde S, Smith A, Wang X, Sasangohar F. Effects of COVID-19 on college students’ mental health in the United States: Interview survey study. J Med Internet Res. (2020) 22:e21279. doi: 10.2196/21279
Bruno G, Panzeri A, Granziol U, Alivernini F, Chirico A, Galli F, Lucidi F, Spoto A, Vidotto G, Bertamini M. The Italian COVID-19 psychological research consortium (IT C19PRC): general overview and replication of the UK study. Journal of clinical medicine. 2021 Jan;10(1):52. https://doi.org/10.3390/jcm10010052
Celia, G., Tessitore, F., Cavicchiolo, E., Girelli, L., Limone, P., & Cozzolino, M. (2022). Improving University Students’ Mental Health During the COVID-19 Pandemic: Evidence From an Online Counseling Intervention in Italy. Frontiers in Psychiatry, 13. https://doi.org/10.3389/fpsyt.2022.886538
Wathelet M, Duhem S, Vaiva G, Baubet T, Habran E, Veerapa E, et al. Factors associated with mental health disorders among university students in France confined during the COVID-19 pandemic. JAMA Network Open. (2020) 3:e2025591. doi: 10.1001/jamanetworkopen.2020.25591
Gritsenko V, Skugarevsky O, Konstantinov V, Khamenka N, Marinova T, Reznik A, et al. 19 fear, stress, anxiety, and substance use among Russian and Belarusian university students. Int J Ment Health Addict. (2021) 19:2362– 8. doi: 10.1007/s11469-020-00330-z
Savage MJ, James R, Magistro D, Donaldson J, Healy LC, Nevill M, Hennis PJ. Mental health and movement behaviour during the COVID-19 pandemic in UK university students: Prospective cohort study. Mental Health and Physical Activity. 2020 Oct 1;19:100357. https://doi.org/10.1016/j.mhpa.2020.100357
MATERIALS AND METHODS
8) Please clarify what it means “professional online survey”.
9) I would also recommend to add some information about the recruitment process. How many Universities have been involved? What is the difference between “key” and “common” Universities? When were the students recruited? During online lessons?
10) I also recommend the Authors to move the data about the sample reported in the Results section in the Sample section (from line 149 to line 151).
11) I also recommend the Authors to specify how they reach the number of 26 items (i.e. they sum the number of the items for each instrument) at row 72.
12) In the description of the FCV-19S I suggest to clarify what the rating of the Likert scale means, so 1 corresponds to… and 5 corresponds to…
13) Please provide the Cronbach’s alpha also for the PHQ-9 and the GAD-7.
ANALYSIS
14) Line 139: Please revise this sentence “An RMSEA value less than .05 correlates with a “good” fit…” I would suggest to change “correlates”.
15) Line 143: Please change the verb “confirmed” in the following sentence: “The convergent validity of the C-FV-19S was confirmed through correlation analysis”. I would recommend to use “confirmed” in the Results section.
16) Why the Authors have used in the analysis the PHQ-9 and the GAD-7 as categorical variables and not continuous? Please clarify.
17) Please add information about the estimator used in the CFA.
18) Another important issue for the validation of the scale can be its invariance across genders. I would suggest the Authors to evaluate if the C-FV-19S is full invariant across males and females.
RESULTS
19) Please add the descriptive statistics (included skewness and kurtosis) of all the variables included in the study.
20) It is unclear to me why the Authors decided to remove item 7 and 8. Further information are needed.
21) Line 191-192: I would rephrase the following sentence and I would add the fit indices values also in the text. “The model 191 fit of the single-factor model (Model 1) was unqualified.”
22) The correlation between the two latent factors is quite high: have the Authors thought to add a second-order factor to the model?
DISCUSSION
23) Line 237: Why the Authors have stated that the C-FCV-19S was a “practical” scale? Please clarify.
24) Line 240-241: The Authors stated that “The 240 scores of C- FCV-19S may predict anxiety and depressive symptoms.” However, they conducted a cross-sectional study and no causal assumption can be made. Please clarify the evidence that support this statement.
25) I don’t understand why the Authors chose to describe the various validation study conducted in other contexts in the Discussion section. I would suggest to move this information in the Introduction.
26) Line 258-261: I think that the Authors have raised an important point. Other studies have pointed out that young people with a higher level of moral disengagement are less likely to adopt prevention behaviors during public emergencies such as the COVID-19 pandemic (e.g. Bavel et al., 2020; Alivernini et al., 2021). Moreover, it seems that in young people higher levels of perceived threat concerning personal health or the health of people they care did not correspond to a greater probability to adopt preventive behaviors (Cavicchiolo et al., 2021). I would recommend the Authors to discuss this point further. Please see the references below.
Alivernini, F., Manganelli, S., Girelli, L., Cozzolino, M., Lucidi, F., & Cavicchiolo, E. (2021). Physical Distancing Behavior: The Role of Emotions, Personality, Motivations, and Moral Decision-Making. Journal of Pediatric Psychology, 46(1), 15-26. https://doi.org/10.1093/jpepsy/jsaa122
Bavel, J. J. V., Baicker, K., Boggio, P. S., Capraro, V., Cichocka, A., Cikara, M., Crockett, M. J., Crum, A. J., Douglas, K. M., Druckman, J. N., Drury, J., Dube, O., Ellemers, N., Finkel, E. J., Fowler, J. H., Gelfand, M., Han, S., Haslam, S. A., Jetten, J., ... Willer, R. (2020). Using social and behavioural science to support COVID- 19 pandemic response. Nature Human Behaviour, 4(5), 460–471. 10.1038/s41562-020-0884-z
Cavicchiolo, E., Manganelli, S., Girelli, L., Cozzolino, M., Lucidi, F., & Alivernini, F. (2021). Adolescents at a Distance. European Journal of Health Psychology, 1-10. https://doi.org/10.1027/2512-8442/a000083
27) Line 267: There is a typo “First, an EPA”
28) Line 272: Please add a reference for the following sentence “Some studies have suggested that this was acceptable in large sample studies (n>1000).”
29) Line 280-281: the sentence is repeated twice.
APPENDIX A:
30) Item 4: there is a typo “termina”.
Author Response
Response to Reviewer 4 Comments
Dear Editor,
Thank you for giving me the opportunity to review the paper entitled "Cross-cultural Adaptation and Validation of the Fear of COVID-19 Scale for Chinese University Students: A cross-sectional study". I appreciated the content and originality of the contribution and I think it can be suitable for the publication to the Journal. Nevertheless, I have some concerns that I list below:
Point 1: ABSTRACT: I found this sentence unclear, please revise. “However, the scale of further cross-cultural adaptation and validation in university students is needed.”
Response 1:
Thank you very much for your careful review. We have revised the sentence in the abstract, showing as follows: “However, the validity and reliability of the Fear of COVID-19 Scale have not been fully investigated in Chinese university student groups”.
Point 2: ABSTRACT: It would be helpful to add in the Abstract some information about the factorial structure of the scale (i.e. 2 dimensions, 8 items each).
Response 2:
Thank you very much for your careful review. We have revised the sentence in the abstract, showing as follows: “Using the exploratory factor analysis (EFA), we examined the construct reliability (KMO = 0.920).The confirmatory factor analysis (CFA) confirmed that the bifactor model of scale (including general factor, factor1: the awareness of COVID-19 and physiological arousal, factor 2: fear-related thinking) had a good fit index (χ2 /df =6.18, RMSEA= .067, SRMR =.028, GFI =.986, TLI =.970 and CFI=.988)”.
Point 3: INTRODUCTION I think that the Introduction gives a concise description about the opportunity to validate the FCV-19S in the Chinese context. However, I believe that some other data are needed.
Response 3:
Thank you very much for your excellent suggestion. We have added some context on COVID-19 in China to illustrate the importance of verifying FCV-19s in the Chinese college student population in the section Introduction on Page 2 Lines 66-85, showing as follows:
In China, approximately 80,000 individuals have been diagnosed with COVID-19, with over 4600 officially recorded deaths (Chinese National Health Commission 2020). The massive infectious public health event has put enormous pressure on the Chinese government, health care providers, and the public [18]. Level 1 public health response was activated in 31 Chinese provinces [19]. There are 33.66 million college students nationwide, including 8.83 million inter-provincial students. The continuous spread of the epidemic, strict isolation measures and delays in starting schools, colleges, and universities across the country were expected to influence the mental health of college students [20,21]. Existing studies have found that university students were more vulnerable to the harmful effects of media information overload of the COVID-19 outbreak, including panic, anxiety, and depression [22-24]. Incidences of anxiety and depression among Chinese university students were up to 40%-50% during the COVID-19 epidemic [1, 25-27]. One of the responsibilities of universities is to protect the physical and mental health of students and prevent the possible consequences of the spread of the epidemic [28]. During the COVID-19 epidemic, the National Health Commission of China issued a number of measures to reduce the spread of the virus, such as lockdowns, quarantine, and online teaching, and implemented emergency psychological crisis intervention for the public [29]. Meantime, the Ministry of Education of China has issued guidelines on mental health services in universities, including screening, monitoring the mental health status, and increasing the number of full-time and part-time psychological counselors [30].
Point 4: The Authors should describe in more detail the original version of the scale (how many items, where it was developed, what are its psychometric properties, etc.) and the diverse adaptations. It could be also useful to add a table that summarize the different characteristics and properties of the diverse adaptations of the scale.
Response 4:
Thank you very much for your excellent suggestion. We have added some detail the original version of the scale in the section of Introduction on Page 2 Lines 49-62, showing as follows:
The Fear of COVID-19 Scale (FCV-19S) was developed in an Iranian context in 2020 and shown to have strong reliability and validity scale for assessing fears related to the coronavirus [7]. The final version of FCV-19S was a single-dimensional scale with 7 items and was shown to be significantly correlated with depression and anxiety, making it helpful for identifying these comorbid disorders [7-9]. Subsequently, the FCV-19S has been translated into eighteen different languages [8]. Most of these studies showed that it is a unidimensional scale. However, studies proposed a two-factor structure, such as in Israeli sample [10], Ecuadorian sample [11], Chinese population sample [12] and Russian adolescents [13]. These inconsistent results also show that it is an unstable factor structure of the FCV-19 Scale[14]. Meantime, most studies were small or middle-aged samples. These included work in Iranian (N=717, mean age: 31)[7], Italian (N=249, mean age: 34 ) [14], Saudi (N=639, mean age: 35) populations [15]. These discrepancies may be due to differences in sample characteristics, cultural backgrounds, or experiences of the COVID-19 epidemic, including different, countries, ethnic groups, epidemic control situations, and so on.
Point 5: The Authors should also add a brief paragraph about Chinese context in which the study was conducted both in terms of the spread of the COVID-19 and of the policy adopted by the Authorities (e.g. restrictive policies to free movement of people, hygienic recommendations, etc). This could help the reader to understand the cultural, social and political background of the study.
Response 5:
Thank you very much for your excellent suggestion. We have added some context on the COVID-19 pandemic in China in the third paragraph of the Introduction in the revised manuscript, showing as follows:
In China, approximately 80,000 individuals have been diagnosed with COVID-19, with over 4600 officially recorded deaths (Chinese National Health Commission 2020). The massive infectious public health event has put enormous pressure on the Chinese government, health care providers, and the public [18]. Level 1 public health response was activated in 31 Chinese provinces [19]. There are 33.66 million college students nationwide, including 8.83 million inter-provincial students. The continuous spread of the epidemic, strict isolation measures and delays in starting schools, colleges, and universities across the country were expected to influence the mental health of college students [20,21]. Existing studies have found that university students were more vulnerable to the harmful effects of media information overload of the COVID-19 outbreak, including panic, anxiety, and depression [22-24].
During the COVID-19 epidemic, the National Health Commission of China issued a number of measures to reduce the spread of the virus, such as lockdowns, quarantine, and online teaching, and implemented emergency psychological crisis intervention for the public [29]. Meantime, the Ministry of Education of China has issued guidelines on mental health services in universities, including screening, monitoring the mental health status, and increasing the number of full-time and part-time psychological counselors [30].
6) I don’t understand why the Authors stated, “Existing studies have reported a positive correlation between mobile phone use and psychological stress during the COVID-19 epidemic [16, 17].” Why have the Authors introduced the issue of the correlation between mobile phone use and psychological stress during COVID-19? The link is unclear to me.
Response 6:
Thank you very much for your careful review. In our study, it was not the goal of the study to explore the relationship between mobile phone use and psychological stress during the COVID-19 epidemic. Thus, we have deleted the sentence and revised it in the Introduction section on Page 2 Lines 74-76, showing as follows: “Existing studies have found that university students are more vulnerable to the harmful effects of media information overload of the COVID-19 outbreak, including panic, anxiety, and depression”.
7) I think that the Authors should expand the description of the negative psychological effects of COVID for University students. It would be useful to add evidence from other studies conducted in China, as well as in other cultural contexts. Please, see the following articles for a non-exhaustive list.
Cao W, Fang Z, Hou G, Han M, Xu X, Dong J, Zheng J. The psychological impact of the COVID-19 epidemic on college students in China. Psychiatry research. 2020 May 1;287:112934. https://doi.org/10.1016/j.psychres.2020.112934
Ma Z, Zhao J, Li Y, Chen D, Wang T, Zhang Z, et al. Mental health problems and correlates among 746 217 college students during the coronavirus disease 2019 outbreak in China. Epidemiol Psychiatr Sci. (2020) 29:e181. doi: 10.1017/S2045796020000931
Wang C, Zhao H. The impact of COVID-19 on anxiety in Chinese university students. Front Psychol. (2020) 11:1168. doi: 10.3389/fpsyg.2020.01168
Son C, Hegde S, Smith A, Wang X, Sasangohar F. Effects of COVID-19 on college students’ mental health in the United States: Interview survey study. J Med Internet Res. (2020) 22:e21279. doi: 10.2196/21279
Bruno G, Panzeri A, Granziol U, Alivernini F, Chirico A, Galli F, Lucidi F, Spoto A, Vidotto G, Bertamini M. The Italian COVID-19 psychological research consortium (IT C19PRC): general overview and replication of the UK study. Journal of clinical medicine. 2021 Jan;10(1):52. https://doi.org/10.3390/jcm10010052
Celia, G., Tessitore, F., Cavicchiolo, E., Girelli, L., Limone, P., & Cozzolino, M. (2022). Improving University Students’ Mental Health During the COVID-19 Pandemic: Evidence From an Online Counseling Intervention in Italy. Frontiers in Psychiatry, 13. https://doi.org/10.3389/fpsyt.2022.886538
Wathelet M, Duhem S, Vaiva G, Baubet T, Habran E, Veerapa E, et al. Factors associated with mental health disorders among university students in France confined during the COVID-19 pandemic. JAMA Network Open. (2020) 3:e2025591. doi: 10.1001/jamanetworkopen.2020.25591
Gritsenko V, Skugarevsky O, Konstantinov V, Khamenka N, Marinova T, Reznik A, et al. 19 fear, stress, anxiety, and substance use among Russian and Belarusian university students. Int J Ment Health Addict. (2021) 19:2362– 8. doi: 10.1007/s11469-020-00330-z
Savage MJ, James R, Magistro D, Donaldson J, Healy LC, Nevill M, Hennis PJ. Mental health and movement behaviour during the COVID-19 pandemic in UK university students: Prospective cohort study. Mental Health and Physical Activity. 2020 Oct 1;19:100357. https://doi.org/10.1016/j.mhpa.2020.100357
Response 5:
Thank you very much for your excellent suggestion. We have carefully read the latest references you provided and quoted some of them to revise the issue on Page 2 Lines 66-85, showing as follows:
In China, approximately 80,000 individuals have been diagnosed with COVID-19, with over 4600 officially recorded deaths (Chinese National Health Commission 2020). The massive infectious public health event has put enormous pressure on the Chinese government, health care providers, and the public [18]. Level 1 public health response was activated in 31 Chinese provinces [19]. There are 33.66 million college students nationwide, including 8.83 million inter-provincial students. The continuous spread of the epidemic, strict isolation measures and delays in starting schools, colleges, and universities across the country were expected to influence the mental health of college students [20,21]. Existing studies have found that university students were more vulnerable to the harmful effects of media information overload of the COVID-19 outbreak, including panic, anxiety, and depression [22-24]. Incidences of anxiety and depression among Chinese university students were up to 40%-50% during the COVID-19 epidemic [1, 25-27]. One of the responsibilities of universities is to protect the physical and mental health of students and prevent the possible consequences of the spread of the epidemic [28]. During the COVID-19 epidemic, the National Health Commission of China issued a number of measures to reduce the spread of the virus, such as lockdowns, quarantine, and online teaching, and implemented emergency psychological crisis intervention for the public [29]. Meantime, the Ministry of Education of China has issued guidelines on mental health services in universities, including screening, monitoring the mental health status, and increasing the number of full-time and part-time psychological counselors [30].
MATERIALS AND METHODS
Point 8: Please clarify what it means “professional online survey”.
Response 8:
Thank you very much for your careful review. We have revised the sentence on Page 2 Lines 93-95 of the abstract, showing as follows: “ The cross-sectional study was carried out using a professional online survey in southwest China, from October to November 2020. We used an online survey website (www.wjx.cn) to create and distribute the survey”.
Point 9: I would also recommend to add some information about the recruitment process. How many Universities have been involved? What is the difference between “key” and “common” Universities? When were the students recruited? During online lessons?
Response 9:
Thank you very much for your excellent suggestion. We have added the details of the recruitment process on Page 2 Line 95-98 the abstract, showing as follows: “The data was collected in the classroom by investigators from four universities, including two key universities (excellent universities) and two common Universities. A total of 2550 participants were studying in the universities and 2334 of which were valid (effective rate =91%)”.
Point 10: I also recommend the Authors to move the data about the sample reported in the Results section in the Sample section (from line 149 to line 151).
Response 10:
Thank you very much for your excellent suggestion. We have moved the sample characteristics and Table 1 in the Results section into the section of Study design and participant characteristics on Page 3.
Table 1. Participant characteristics (n =2234).
|
Variable |
|
Mean ±standard deviation or n (%) |
|
Age (years old) |
|
19±1.29 |
|
|
17-19 |
1075 (46.1) |
|
|
20-22 |
1198 (51.3) |
|
|
23-29 |
61 (2.6) |
|
University |
|
|
|
|
key university |
965(41.3) |
|
|
common university |
1369(58.7) |
|
Gender |
|
|
|
|
Male |
1275 (54.6) |
|
|
female |
1059 (45.4) |
|
PHQ-9 |
|
4.88±4.82 |
|
|
Negative-depression |
1850 (79.3) |
|
|
Positive-depression |
484 (20.7) |
|
GAD-7 |
|
3.31±3.98 |
|
|
Negative-anxiety |
2132 (91.3) |
|
|
Positive-anxiety |
202 (8.7) |
|
C-FV-19S |
|
16.04±6.12 (8-50) |
Point 11: I also recommend the Authors to specify how they reach the number of 26 items (i.e. they sum the number of the items for each instrument) at row 72.
Response 11:
Thank you very much for your excellent suggestion. We have revised the sentence on Page 3 Lines 100-101, showing as follows: “The questionnaire included 26 items (they sum the number of the items for each instrument), thus meeting the ideal sample size”.
Point 12: In the description of the FCV-19S I suggest to clarify what the rating of the Likert scale means, so 1 corresponds to… and 5 corresponds to…
Response 12:
Thank you very much for your excellent suggestion. We have revised the sentence on Page 4 Line 121-123, showing as follows: “It was later revised into a 7-item unidimensional scale with a five-point Likert rating scale, with 1 corresponding to strongly disagree and 5 corresponding to strongly disagree 5.
Point 13: Please provide the Cronbach’s alpha also for the PHQ-9 and the GAD-7.
Response 13:
Thank you very much for your excellent suggestion. We have added the Cronbach’s alpha of the PHQ-9 and GAD-7 on Page 4, showing as follows: “In this study, the Cronbach's α of GAD-7 in this sample was 0.923”, and “In this study, the Cronbach's α of GAD-7 in this sample was 0.923.”
ANALYSIS
Point 14: Line 139: Please revise this sentence “An RMSEA value less than .05 correlates with a “good” fit…” I would suggest to change “correlates”.
Response 14:
Thank you very much for your excellent suggestion. We have revised the sentence on Page X Line X, showing as follows: “An RMSEA value is less than .05 or 0.08, the model can be assumed to demonstrate a “good” fit or “acceptable” fit to the data, respectively.”
Point 15: Line 143: Please change the verb “confirmed” in the following sentence: “The convergent validity of the C-FV-19S was confirmed through correlation analysis”. I would recommend to use “confirmed” in the Results section.
Response 14:
Thank you very much for your excellent suggestion. We have revised the sentence on Page X Line X, showing as follows: “The convergent validity of the C-FV-19S was examined through correlation analysis of the C-FV-19S and the PHQ-9 and GAD-7”.
Point 16: Why the Authors have used in the analysis the PHQ-9 and the GAD-7 as categorical variables and not continuous? Please clarify.
Response 16:
Thank you very much for your careful review. I will explain and clarify this issue as follows: Firstly, the PHQ-9 and the GAD-7 were regarded as continuous variables, when we analyzed the correlation between the C-FV-19S and PHQ-9 or the GAD-7. Second, the PHQ-9 and the GAD-7 were regarded as categorical variables, when we used the receiver operating characteristic curve (ROC) analysis to gain the cut-off value of the C-FV-19S by using the anxiety-positive and depression positive.
Point 17:Please add information about the estimator used in the CFA.
Response 17:
Thank you very much for your excellent suggestion. We have added the information on Page X Line X, showing as follows: “We used the Maximum Likelihood Estimates in the CFA and ran CFA on sample 2”.
Point 18:Another important issue for the validation of the scale can be its invariance across genders. I would suggest the Authors to evaluate if the C-FV-19S is full invariant across males and females.
Response 17:
Thank you very much for your excellent suggestion. We have referred to some references for statistical analysis and examined the models in females and males by using the SEM, respectively, and got consistent results, shown as follows:
References:
[1] Leue A , Beauducel A . The PANAS structure revisited: on the validity of a bifactor model in community and forensic samples.[J]. Psychological Assessment, 2011, 23(1):215-25. 10.1037/a0021400
[2] Torres-Vallejos J , Juarros-Basterretxea J , Oyanedel J C , et al. A Bifactor Model of Subjective Well-Being at Personal, Community, and Country Levels: A Case With Three Latin-American Countries[J]. Frontiers in Psychology, 2021, 12. 10.3389/fpsyg.2021.641641
[3] Di M , Jia N , Wang Q , et al. A bifactor model of the Wong and Law Emotional Intelligence Scale and its association with subjective well-being[J]. The Journal of Positive Psychology, 2020(6):1-12. 10.1080/17439760.2020.1791947
RESULTS
Point 19:Please add the descriptive statistics (included skewness and kurtosis) of all the variables included in the study.
Response 19:
Thank you very much for your excellent suggestion. We have added the descriptive statistics (including skewness and kurtosis) of all the variables in revised Table 2.
Table 2. Item Analysis of the Chinese version Fear of COVID-19 scale (The C-FCV-19S).
|
The C-FCV-19S items |
Mean (SD) |
Skewness |
Kurtosis |
correlation |
p-value |
|
1. I am most afraid of corona virus-19. |
3.02(1.33) |
-.037 |
-1.134 |
0.678 |
<0.001 |
|
2. It makes me uncomfortable to think about coronavirus-19. |
2.61(1.24) |
.314 |
-.888 |
0.783 |
<0.001 |
|
3. I worry a lot about coronavirus-19. |
2.30(1.15) |
.577 |
-.500 |
0.771 |
<0.001 |
|
4. Coronavirus-19 is almost always terminal. |
1.61(.93) |
1.503 |
1.671 |
0.64 |
<0.001 |
|
5. Coronavirus-19 is an unpredictable disease. |
2.36(1.28) |
.551 |
-.786 |
0.636 |
<0.001 |
|
6. My hands become clammy when I think about coronavirus-19. |
1.60(.936) |
1.646 |
2.274 |
0.772 |
<0.001 |
|
7. I am afraid of losing my life because of coronavirus-19. |
2.15(1.30) |
.858 |
-.424 |
0.747 |
<0.001 |
|
8. When watching news and stories about coronavirus-19 on social media, I become nervous or anxious. |
2.28(1.18) |
.562 |
-.680 |
0.765 |
<0.001 |
|
9. I cannot sleep because I’m worrying about getting coronavirus-19. |
1.46(.83) |
1.895 |
3.225 |
0.715 |
<0.001 |
|
10. My heart races or palpitates when I think about getting coronavirus-19. |
1.60(.97) |
1.660 |
2.116 |
0.741 |
<0.001 |
Point 20:It is unclear to me why the Authors decided to remove item 7 and 8. Further information are needed.
Response 20:
Thank you very much for your excellent suggestion. We have revised these sentences on Page 6 Lines 212-213, showing as follows: “The factor loadings of items 7 and 8 exceeded 0.5 on both Factor 1 and Factor 2. Thus, these two items were excluded, leaving eight items (see Appendix)”.
Point 21:Line 191-192: I would rephrase the following sentence and I would add the fit indices values also in the text. “The model 191 fit of the single-factor model (Model 1) was unqualified.”
Response 21:
Thank you very much for your excellent suggestion. We have revised these sentences on Page 7 Lines 222-226, showing as follows: “Model 1 was a unidimensional model, Model 2 was a two-factor model, and Model 3 was a bifactor model. The fit indices of the two models are shown in Table 4. The model fit of the single-factor model (Model 1) was unqualified as its RMSEA values (.229) and χ2 /df (61.95), the SRMR (.1002) was higher than the ideal value (≤0.08) and the GFI (.754), TLI (.946), and CFI (.747) indices were also lower than the ideal value (.90)”.
Point 22: The correlation between the two latent factors is quite high: have the Authors thought to add a second-order factor to the model?
Response 22:
According to your excellent suggestion, we have analyzed the SEM again, by adding a second-order factor to the model. Unfortunately, the second-order factor model wasn’t identified. Then, we speculated there should be a hidden variable (a general factor). Further, we explore the bifactor model, and the fit indices of the bifactor model were obtained to the ideal values (χ2 /df =6.18, RMSEA=.067, SRMR =.028, GFI =.986, TLI =.970 and CFI=.988). Therefore, we reported the bifactor model (Model 3) in the revised manuscript.
Table 4. Fit indices of models in the CFA of the Fear of COVID-19 Scale in test sample 2
|
|
c2(df.) |
c 2/df |
RMSEA |
SRMR |
GFI |
CFI |
TLI |
NFI |
RFI |
IFI |
|
Threshold value |
≤5.0 |
≤0.08 |
≤0.08 |
>0.09 |
>0.09 |
>0.09 |
>0.09 |
>0.09 |
>0.09 |
|
|
Model 1 |
1239.172(20)*** |
61.95 |
0.229 |
0.1002 |
0.754 |
0.747 |
0.646 |
0.744 |
0.642 |
0.748 |
|
Model 2 |
450.63(19)*** |
23.68 |
0.14 |
0.0774 |
0.907 |
0.936 |
0.906 |
0.934 |
0.902 |
0.936 |
|
Model 3 |
68.055(11)*** |
6.18 |
0.067 |
0.028 |
0.986 |
0.988 |
0.970 |
0.986 |
0.964 |
0.988 |
***Indicates significance at the 0.001 level. Model1: single-factor model; Model 2: two factors model consisting of factor 1 and factor 2. Model 3: bifactor model.
DISCUSSION
Point 23: Line 237: Why the Authors have stated that the C-FCV-19S was a “practical” scale? Please clarify.
Response 23:
Thank you very much for your excellent suggestion. We have revised these sentences on Page 9 Lines 278-280, showing as follows: “This study assessed the cultural adaptation of the C-FCV-19S and found it to be psychometrically valid for Chinese university students and to help quickly screen the fear of COVID-19”.
Point 24: Line 240-241: The Authors stated that “The 240 scores of C- FCV-19S may predict anxiety and depressive symptoms.” However, they conducted a cross-sectional study and no causal assumption can be made. Please clarify the evidence that support this statement.
Response 24:
Thank you very much for your careful review. We have revised these sentences on Page 9 Lines 282-283, showing as follows: “The C-FCV-19S score was a positive association with anxiety and depressive symptoms”.
Point 25: I don’t understand why the Authors chose to describe the various validation study conducted in other contexts in the Discussion section. I would suggest to move this information in the Introduction.
Response 25:
Thank you very much for your excellent suggestion. We have moved these sentences into the section of introduction on Page 2 Lines 58-62, showing as follows: “Meantime, most studies were small or middle-aged samples. These included work in Iranian (N=717, mean age: 31), Italian (N=249, mean age: 34), Saudi (N=639, mean age: 35) populations. These discrepancies may be due to differences in sample characteristics, cultural backgrounds, or experiences of the COVID-19 epidemic, including different, countries, ethnic groups, epidemic control situations, and so on”.
Point 26: Line 258-261: I think that the Authors have raised an important point. Other studies have pointed out that young people with a higher level of moral disengagement are less likely to adopt prevention behaviors during public emergencies such as the COVID-19 pandemic (e.g. Bavel et al., 2020; Alivernini et al., 2021). Moreover, it seems that in young people higher levels of perceived threat concerning personal health or the health of people they care did not correspond to a greater probability to adopt preventive behaviors (Cavicchiolo et al., 2021). I would recommend the Authors to discuss this point further. Please see the references below.
Response 26:
Thank you very much for your excellent suggestion and for providing these references. We have added the discussion of the issue on Page 10 Lines 303-315, showing as follows: “More importantly, this difference may come from the specificity of college students as a group. It has been illustrated that young people with higher levels of moral disengagement are less likely to engage in preventive behaviors in public crisis situations such as the COVID-19 pandemic [50,51]. In addition, 7 and 8 items of the FCV-19S belong to threat perception. It seems that in young people higher levels of perceived threats concerning personal health or the health of loved ones did not correspond to a greater probability to adopt preventive behaviors [52]. One possible explanation is that adolescents are at a much lower risk of contracting the most severe symptoms of COVID-19 [53] and therefore they consider the probability of being infected with serious harm to be very low. In other words, threat perception does not influence the adoption of healthy behaviors by adolescents in the face of the COVID-19 epidemic [54]. Taken together, we have good reason to speculate that two items are excluded from the CFA analysis”.
References :
[51]Alivernini, F., Manganelli, S., Girelli, L., Cozzolino, M., Lucidi, F., & Cavicchiolo, E. (2021). Physical Distancing Behavior: The Role of Emotions, Personality, Motivations, and Moral Decision-Making. Journal of Pediatric Psychology, 46(1), 15-26. https://doi.org/10.1093/jpepsy/jsaa122
[52]Bavel, J. J. V., Baicker, K., Boggio, P. S., Capraro, V., Cichocka, A., Cikara, M., Crockett, M. J., Crum, A. J., Douglas, K. M., Druckman, J. N., Drury, J., Dube, O., Ellemers, N., Finkel, E. J., Fowler, J. H., Gelfand, M., Han, S., Haslam, S. A., Jetten, J., ... Willer, R. (2020). Using social and behavioural science to support COVID- 19 pandemic response. Nature Human Behaviour, 4(5), 460–471. 10.1038/s41562-020-0884-z
[53]Cavicchiolo, E., Manganelli, S., Girelli, L., Cozzolino, M., Lucidi, F., & Alivernini, F. (2021). Adolescents at a Distance. European Journal of Health Psychology, 1-10. https://doi.org/10.1027/2512-8442/a000083
Point 27: Line 267: There is a typo “First, an EPA”
Response 27:
Thank you very much for your careful review. We have revised the sentence on Page10 Line 317, showing as follows: “Firstly, EPA found that two factors of the C-FCV-19S, based on factors with eigenvalues, were greater than those of random datasets”.
Point 28: Line 272: Please add a reference for the following sentence “Some studies have suggested that this was acceptable in large sample studies (n>1000).”
Response 28:
Thank you very much for your excellent suggestion. We have added two references for the sentence, showing as follows:
- Hu, L.; Bentler, P. M.. Cutoff criteria for fit indexes in covariance structure analysis: Conventional criteria versus new alternatives. Sturctural Equation Modeling, 1999,6, 1-55.
- Wen, Zhonglin; Hau Kit-Tai; Herber, W.Marsh. Structural equation model testing: cutoff criteria for goodness of fit indices and chi-square test. Acta Psychologica Sinica,2004,36(2):186-194.
Point 29: Line 280-281: the sentence is repeated twice.
Response 29:
Thank you very much for your careful review. We have revised the issue.
APPENDIX A:
Point 30: Item 4: there is a typo “termina”.
Response 30:
Thank you very much for your careful review. We have revised these sentences on append, showing as follows: “Coronavirus-19 is almost always terminal”.
This manuscript is a resubmission of an earlier submission. The following is a list of the peer review reports and author responses from that submission.